

# Fermion mass hierarchies
# from supersymmetric gauged flavour symmetry in 5D

Ketan M. Patel⋆

Physical Research Laboratory, Navarangpura, Ahmedabad-380 009, India

⋆ kmpatel@prl.res.in

## Abstract

A mechanism to generate realistic fermion mass hierarchies based on supersymmetric gauged $U(1)_F$ symmetry in flat five-dimensional (5D) spacetime is proposed. The fifth dimension is compactified on $S^1/Z_2$ orbifold. The standard model fermions charged under the extra abelian symmetry along with their superpartners live in the 5D bulk. Bulk masses of fermions are generated by the vacuum expectation value of $N = 2$ superpartner of $U(1)_F$ gauge field, and they are proportional to $U(1)_F$ charges of respective fermions. This decides localization of fermions in the extra dimension, which in turn gives rise to exponentially suppressed Yukawa couplings in the effective 4D theory. Anomaly cancellation puts stringent constraints on the allowed $U(1)_F$ charges which leads to correlations between the masses of quarks and leptons. We perform an extensive numerical scan and obtain several solutions for anomaly-free $U(1)_F$, which describe the observed pattern of fermion masses and mixing with all the fundamental parameters of order unity. It is found that the possible existence of SM singlet neutrinos substantially improves the spectrum of solutions by offering more freedom in choosing $U(1)_F$ charges. The model predicts $Z'$ boson mediating flavour violating interactions in both the quark and lepton sectors with the couplings which can be explicitly determined from the Yukawa couplings.

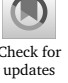

# 1   Introduction

The noteworthy features of the observed pattern of fermion masses and mixings are: (a) the charged fermion masses range over six orders of magnitude, (b) inter-generational hierarchy in the up-type quarks is much stronger than that in the down-type quarks or the charged leptons, (c) neutrinos are mildly hierarchical, (d) the quark mixing angles are hierarchical and small while (e) the lepton mixing angles are of $\mathcal{O}(1)$, see Table 1 for example. The Standard Model (SM) extended with the seesaw mechanism for neutrino masses can accommodate all these empirical observations, but it does not provide any rational and coherent understanding of the above features. This constitutes the so-called flavour puzzle, and many theories have been put forward to address it, see for example [1] for an overview of the subject.

One of the simplest and earliest proposals to address fermion mass hierarchy is the Froggatt-Nielsen (FN) mechanism [2]. Fermions of different generations have different charges under a global $U(1)$ symmetry, breaking of which induces power-suppressed couplings in the effective theory [3, 4]. The underlying $U(1)$ symmetry can also be gauged, however, the set of FN charges required for realistic fermion mass spectrum in these models leads to anomalies and additional fields and/or new mechanisms are required to cancel them [5–11]. Alternatively, models based on extra spatial dimension(s) can also give rise to exponentially suppressed effective couplings by appropriate localisation of the fermions of different generations in the extra dimension [12–14]. The features (a) and (d) mentioned above can naturally be realised in these models without relying on any arbitrarily small or large dimensionless parameters. There exists freedom to choose the FN charges or the bulk mass parameters for different species of fermions in these models which can be used to accommodate the remaining features (b), (c) and (e). More predictive frameworks can be obtained by implementing these mechanisms in the unified models [15, 16] which provide partial or complete unification of the quarks and leptons of a given generation. These constructions provide a platform to understand all the features listed above because of the correlations among the FN charges or bulk masses of various fermions. Several models exploiting this or similar mechanisms have been studied, see for example [17–26].

An interesting framework along this direction is proposed by Kitano and Li in [21] based on supersymmetric gauge theory in flat five dimensional (5D) spacetime with an extra dimension compactified on $S^1/Z_2$ orbifold. The SM fermions and gauge fields along with their superpartners can propagate in the extra dimension while the Higgs fields live on a 4D fixed point. The SM gauge symmetry is embedded in $SO(10)$ grand unified theory which unifies quarks and leptons of a given generation and predicts a common bulk mass for them. The later controls the flavour hierarchies in the effective 4D theory. While quark-lepton unification provides elegant and predictive framework for flavour hierarchies in this setup, it becomes necessary to

Table 1: Quark and lepton masses and mixing angles at 10 TeV. The charged fermion masses and quark mixing angles are taken from [27] while neutrino masses and lepton mixing angles are derived from [28] assuming normal ordering in the neutrino masses.

| | | |
|---|---|---|
| $m_u/m_t = 7.22 \times 10^{-6}$ | $m_c/m_t = 3.52 \times 10^{-3}$ | $m_t = 137.4\,\text{GeV}$ |
| $m_d/m_b = 9.99 \times 10^{-4}$ | $m_s/m_b = 1.98 \times 10^{-2}$ | $m_b = 2.13\,\text{GeV}$ |
| $m_e/m_\tau = 2.79 \times 10^{-4}$ | $m_\mu/m_\tau = 5.88 \times 10^{-2}$ | $m_\tau = 1.80\,\text{GeV}$ |
| $m_{\nu_1}/m_{\nu_3} \in [0,1]$ | $m_{\nu_2}/m_{\nu_3} \in [0.17,1]$ | $m_{\nu_3} \in [0.05, 0.1]\,\text{eV}$ |
| $\theta_{12}^q = 0.2274$ | $\theta_{23}^q = 0.04364$ | $\theta_{13}^q = 0.00377$ |
| $\theta_{12}^l = 0.5558$ | $\theta_{23}^l = 0.7788$ | $\theta_{13}^l = 0.1487$ |

break the unified gauge symmetry in the bulk to obtain realistic spectrum of fermion masses and mixing angles [22, 23].

In this paper, we propose an alternative framework which does not use the premise of quark-lepton unification but still offers a predictive setup to address the flavour puzzle. The framework is based on an additional gauged flavour symmetry, $U(1)_F$, constructed on supersymmetric 5D orbifold. The supersymmetry (SUSY) and gauge invariance allow only gauge interactions in bulk. A vacuum expectation value (VEV) of $N = 2$ superpartner of the $U(1)_F$ vector multiplet generates bulk masses for various fermions proportional to their $U(1)_F$ charges which decide the adequate strength of their couplings with the Higgs localised on the SM brane. More importantly, the $U(1)_F$ charges of various fermions are constrained from the requirement of anomaly-free 5D theory. Anomaly cancellation gives rise to inter-generational as well as inter-species correlations between the $U(1)_F$ charges of various fermions and predicts relations between the hierarchies of quarks and leptons even in the absence of quark-lepton unification. By analysing these correlations analytically and numerically, we give an example set of $U(1)_F$ charges and discuss their viability in explaining the features (a) to (e) listed above.

We discuss the basic construction of supersymmetric $U(1)$ on 5D orbifold in the next section. The effective SM Yukawa couplings obtained from full 5D theory is discussed in section 3. In section 4, we analytically discuss some examples of the anomaly-free choice of $U(1)_F$ charges and their consequences on fermion mass hierarchies. A comprehensive numerical search for realistic flavour spectrum has been performed, and relevant results are given in section 5. In section 6, we discuss some phenomenological implications of the underlying framework and summarize in section 7. We also give an explicit solution in Appendix A.

## 2   Supersymmetric $U(1)$ on $S^1/Z_2$

We briefly review $N = 1$ supersymmetric abelian gauge theory constructed in five-dimensional flat spacetime [29]. The extra dimension is compactified on $S^1/Z_2$. It is convenient to discuss the spectrum and interactions of this theory in terms of $N = 2$ superspace formalism [30]. In this language, a 5D $N = 1$ vector multiplet can be decomposed into a chiral superfield $\chi$ and a vector superfield $\mathcal{V}$. Similarly, a 5D $N = 1$ hypermultiplet contains a pair of 4D $N = 1$ chiral superfields, $\mathcal{F}$ and $\mathcal{F}^c$. All the superfields are periodic under $y \to y + 2\pi R$ where $y$ denotes the coordinate of the fifth dimension, and $R$ is the radius of $S^1$. Under $Z_2$ parity, $\chi(x^\mu, -y) = -\chi(x^\mu, y)$ and $\mathcal{F}^c(x^\mu, -y) = -\mathcal{F}^c(x^\mu, y)$ while the other fields are assumed to remain even. The gauge and SUSY invariant 5D action involving vector and hyper multiplets

can be written in terms of the decomposed superfields as [30, 31]

$$
\begin{aligned}
S_{5D} &= \int_0^{\pi R} dy \int d^4x \left[ \frac{1}{4} \int (d^2\theta \, W^\alpha W_\alpha + \text{h.c.}) + \int d^4\theta \left( \partial_y \mathcal{V} - \frac{1}{\sqrt{2}} (\chi + \overline{\chi}) \right)^2 \right. \\
&\quad + \left. \int d^4\theta \left( \overline{\mathcal{F}} e^{2g_5 q \mathcal{V}} \mathcal{F} + \overline{\mathcal{F}^c} e^{-2g_5 q \mathcal{V}} \mathcal{F}^c \right) + \left( \int d^2\theta \, \mathcal{F}^c \left( \partial_y - \sqrt{2} g_5 q \chi \right) \mathcal{F} + \text{h.c.} \right) \right] . \quad (1)
\end{aligned}
$$

Here, $g_5$ is the $U(1)$ gauge coupling constant, $q$ is $U(1)$ charge of chiral multiplet $\mathcal{F}$ and $W^\alpha$ is a the field strength. In a more general construction, it is also possible to introduce a $y$ dependent kink mass term, $m(y) = m \, \text{sgn}(y)$, for $\mathcal{F}, \mathcal{F}^c$ and/or similar $Z_2$ odd Fayet-Iliopoulos term [32, 33] for the $U(1)$ gauge field in Eq. (1). However, we do not consider these terms in the present work. Their vanishing value is protected by $Z_2$ parity. With this, the theory described by $S_{5D}$ contains only the gauge interaction characterised by single parameter $g_5$.

The 4D spectrum of the theory can be obtained by minimizing the variation of $S_{5D}$ and using Kaluza-Klein (KK) expansion of the bulk superfields. The boundary conditions imposed by $Z_2$ parity allow existence of massless modes for only $\mathcal{V}$ and $\mathcal{F}$ on the fixed points. In this way, the compactification breaks $N = 2$ SUSY down to $N = 1$ in the 4D theory. If the scalar component of $\chi$ acquires a vacuum expectation value (VEV), it generates kink mass term for $\mathcal{F}, \mathcal{F}^c$. Explicitly, using the KK expansion $\mathcal{F}(x^\mu, y) = \sum_n F_n(x^\mu) f_n(y)$, $\mathcal{F}^c(x^\mu, y) = \sum_n F_n^c(x^\mu) f_n^c(y)$ and the matching condition

$$
\int_0^{\pi R} dy \int d^4x \int d^2\theta \, \mathcal{F}^c \left( \partial_y - \sqrt{2} g_5 q \chi \right) \mathcal{F} = \int d^4x \int d^2\theta \sum_n m_n F_n^c F_n , \quad (2)
$$

one finds following equations for the profile functions:

$$
\left( \partial_y - m \right) f_n(y) = m_n f_n^c(y), \quad \left( \partial_y + m \right) f_n^c(y) = -m_n f_n(y), \quad (3)
$$

where $m \equiv \sqrt{2} g_5 q \langle \chi \rangle$ and $m_n$ are masses of the 4D modes of chiral superfields. The above equations along with the normalization condition

$$
\int_0^{\pi R} dy \, f_n(y) f_m(y) = \delta_{mn} , \quad (4)
$$

give rise to the following wavefunction profile for the massless mode $F_0$:

$$
f_0(y) = \sqrt{\frac{2m}{e^{2m\pi R} - 1}} \, e^{my} . \quad (5)
$$

As a result, the massless mode can be localised on $y = 0$ brane for $m < 0$ and on $y = \pi R$ brane for $m > 0$. For $m = 0$, the profile is constant in the fifth dimension. This result is the most relevant feature of the underlying framework which will be used to generate hierarchical couplings for the SM fermions. There also exists a massless mode of the vector superfield $\mathcal{V}$ with a flat wave-function given by $(\pi R)^{-1/2}$. The effective 4D gauge coupling of $U(1)$ is thus given by $g_4 = g_5/\sqrt{\pi R}$. The other KK modes of vector and various chiral superfields are massive and heavier than the compactification scale $R^{-1}$.

Anomaly of $U(1)$ gauge theory compactified on $S^1/Z_2$ is discussed in [33–35]. In the absence of hypermultiplet, the theory is anomaly-free as the chiral superfield $\chi$ is chargeless under $U(1)$. Computation of anomaly in the presence of hypermultiplets charged under $U(1)$ implies [34]

$$
\partial_M J^M = \frac{1}{2} (\delta(y) + \delta(y - \pi R)) \mathcal{Q} , \quad (6)
$$

where $J^M$ is 5D current and

$$\mathcal{Q} = \frac{g_4^2}{16\pi^2} \operatorname{tr} q\, F \cdot \tilde{F}, \tag{7}$$

is the usual 4D chiral anomaly of Dirac fermions interacting with gauge potential. Consequently, the anomaly of the full theory is completely localised on the fixed points, and it does not depend on the details of the bulk parameters. Therefore, it is sufficient to eliminate the anomaly of the 4D effective theory in order to ensure anomaly-free 5D theory. More specifically, if the theory contains a set of 5D $N = 1$ hypermultiplets, all it is required to cancel the anomaly is that the $n = 0$ modes of $\mathcal{F}$ constitute an anomaly-free content of the effective 4D theory. This gives rise to an important constraint on the massless spectrum of the theory and on the choice of $U(1)$ charges.

## 3  Standard Model Yukawa couplings from $U(1)_F$

We now implement the above framework in the standard model. The SM gauge group is extended to include $U(1)_F$ as an additional gauged flavour symmetry. We assume that the SM fermions charged under $U(1)_F$ and their superpartners live in the fifth dimension while the Higgs sector is localized on one of the 4D fixed points which we choose as $y = 0$ without loss of generality. Orbifold compactification leaves $N = 1$ supersymmetry unbroken on the fixed points, and therefore we discuss the 4D effective theory in the formalism of the Minimal Supersymmetric Standard Model (MSSM). The remaining SUSY in 4D theory can be broken softly in a usual way [36].

Following the discussion in the previous section, the 5D hypermultiplet can be generalised to include three generations of quarks and leptons superfields such that $\mathcal{F} = \mathcal{Q}_i, \mathcal{U}_i^c, \mathcal{D}_i^c, \mathcal{L}_i, \mathcal{E}_i^c$, with $i = 1, 2, 3$. The MSSM Higgs superfields $H_u$, $H_d$ live on the SM ($y = 0$) brane. The 5D superpotential characterizing Yukawa interactions in the underlying framework can be written as

$$W_{5D} = \frac{\delta(y)}{\Lambda} \left( (\mathcal{Y}_u)_{ij}\, \mathcal{Q}_i \mathcal{U}_j^c H_u + (\mathcal{Y}_d)_{ij}\, \mathcal{Q}_i \mathcal{D}_j^c H_d + (\mathcal{Y}_e)_{ij}\, \mathcal{L}_i \mathcal{E}_j^c H_d \right), \tag{8}$$

where $\Lambda$ is a cut-off scale and $\mathcal{Y}_{u,d,e}$ are matrices consist of dimensionless couplings of approximately similar magnitude. Note that these couplings do not respect $U(1)_F$ symmetry in general. They can arise from the VEVs of flavon fields which break $U(1)_F$ symmetry on $y = 0$ brane. The MSSM matter spectrum arise from the zero modes of various superfields in $\mathcal{F}$. Performing KK expansion and integrating over fifth dimension, the above $W_{5D}$ results into the following effective 4D superpotential involving the massless modes of quark and lepton superfields:

$$W_{4D} = (Y_u)_{ij} Q_i U_j^c H_u + (Y_d)_{ij} Q_i D_j^c H_d + (Y_e)_{ij} L_i E_j^c H_d + ..., \tag{9}$$

where ellipses denote interactions involving massive KK modes. Using KK expansions and Eq. (5), the Yukawa coupling matrices $Y_{u,d,e}$ can be obtained by matching $W_{5D}$ and $W_{4D}$ as

$$Y_u = \frac{M_c}{\Lambda} \xi_Q \mathcal{Y}_u \xi_{U^c}, \quad Y_d = \frac{M_c}{\Lambda} \xi_Q \mathcal{Y}_d \xi_{D^c}, \quad Y_e = \frac{M_c}{\Lambda} \xi_L \mathcal{Y}_e \xi_{E^c}, \tag{10}$$

where $M_c = (\pi R)^{-1}$ is compactification scale. The $3 \times 3$ diagonal matrices $\xi_F$, for $F = Q, U^c, D^c, L, E^c$, have $i^{\text{th}}$ diagonal element

$$\xi_{F_i} = \sqrt{\frac{2c\, X_{F_i}}{e^{2c X_{F_i}} - 1}}, \tag{11}$$

where $X_{F_i}$ is $U(1)_F$ charge of $F_i$ and $c = \sqrt{2}g_5\langle\chi\rangle\pi R$ is a dimensionless parameter.

For neutrino masses, we assume a Weinberg operator in $W_{5D}$ which, upon compactification, results into the following effective operator in 4D:

$$W_{4D} \supset \frac{1}{\Lambda'}(Y_\nu)_{ij}L_iL_jH_uH_u\,, \tag{12}$$

where $\Lambda'$ characterizes lepton number violation scale and

$$Y_\nu = \frac{M_c}{\Lambda}\xi_L\,\mathcal{Y}_\nu\,\xi_L\,. \tag{13}$$

Here, $\mathcal{Y}_\nu$ is also $3 \times 3$ matrix with elements of a similar magnitude and they break $U(1)_F$ in general. It is also straight-forward to implement type-I seesaw mechanism as an origin of the Weinberg operator within this framework. However, our discussion on the flavour spectrum does not crucially depend on such detail.

It can be seen from Eqs. (10, 11, 13) that the hierarchical mass spectrum of quarks and leptons can be explained using the appropriate choice of their $U(1)_F$ charges and all the fundamental parameters of $\mathcal{O}(1)$. For example, a choice of charges

$$X_{F_1} > X_{F_2} > 0 \geq X_{F_3}\,, \tag{14}$$

with $c > 0$ localizes the first and second generation fermions away from $y = 0$ brane. This arrangement leads to small masses of the first two generation fermions in comparison to that of the third generation which is localised on the SM brane. The stronger hierarchy in the up-type quark masses and feeble hierarchy in neutrino masses compared to the moderately hierarchical charged leptons and down-type quarks can be obtained using suitable choices for respective $X_F$. However, the requirement of anomaly cancellation severely restricts such possibilities and imply only specific choices for various $X_F$.

## 4  Anomaly cancellation and correlations among fermion mass hierarchies

As discussed in section 2, it is sufficient for an anomaly-free $U(1)_F$ 5D theory to have vanishing anomalies on the 4D fixed points. This in turn restricts the choices for the $U(1)_F$ charges $X_{F_i}$ of the superfields $F = Q, U^c, D^c, L, E^c, N^c$ where we also include three generations of the SM singlet neutrinos in the fermion spectrum. The anomaly cancellation (AC) requirement with one $U(1)$ is comprised of six independent conditions. We reproduce them here in our notation for convenience. The $SU(3)^2 \times U(1)_F$, $SU(2)^2 \times U(1)_F$, $U(1)_Y^2 \times U(1)_F$, $U(1)_Y \times U(1)_F^2$, $U(1)_F^3$ and the gauge-gravity anomaly conditions are respectively given by

$$\sum_{i=1}^{3}\left(2X_{Q_i}+X_{U_i^c}+X_{D_i^c}\right)=0\,, \tag{15}$$

$$\sum_{i=1}^{3}\left(3X_{Q_i}+X_{L_i}\right)=0\,, \tag{16}$$

$$\sum_{i=1}^{3}\left(X_{Q_i}+3X_{L_i}+8X_{U_i^c}+2X_{D_i^c}+6X_{E_i^c}\right)=0\,, \tag{17}$$

$$\sum_{i=1}^{3}\left(X_{Q_i}^2-X_{L_i}^2-2X_{U_i^c}^2+X_{D_i^c}^2+X_{E_i^c}^2\right)=0\,, \tag{18}$$

$$\sum_{i=1}^{3}\left(6X_{Q_i}^3 + 2X_{L_i}^3 + 3X_{U_i^c}^3 + 3X_{D_i^c}^3 + X_{E_i^c}^3 + X_{N_i^c}^3\right) = 0\,, \tag{19}$$

$$\sum_{i=1}^{3}\left(6X_{Q_i} + 2X_{L_i} + 3X_{U_i^c} + 3X_{D_i^c} + X_{E_i^c} + X_{N_i^c}\right) = 0\,. \tag{20}$$

The Right Handed (RH) neutrinos, being the SM gauge singlets, contribute only in the anomalies corresponding to $U(1)_F^3$ and gauge-gravity. We now discuss the correlations among the fermion mass hierarchies as implied by AC in some of the very simplest scenarios.

## 4.1 Without RH neutrinos

We first assume that either RH neutrinos do not exist or they are singlet under $U(1)_F$, hence $X_{N_i} = 0$. AC conditions involving one $U(1)_F$ in the triangle diagrams get satisfied if

$$\text{tr}X_F = 0\,, \tag{21}$$

for all $F = Q, U^c, D^c, L, E^c$. In addition, the $U(1)_F^3$ anomaly can be eliminated if

$$\text{tr}X_F^3 = 0\,. \tag{22}$$

Non-trivial solutions of Eqs. (21,22) are given by

$$X_F = q_F(1, 0, -1)\,, \tag{23}$$

with $q_F > 0$ following the convention, Eq (14). The remaining AC condition, Eq. (18), then can be fulfilled using one of the following identities:

$$
\begin{aligned}
(i) & \quad X_Q = X_{U^c} = X_{D^c} = X_L = X_{E^c}\,, \\
(ii) & \quad X_Q = X_L \ \text{and} \ X_{U^c} = X_{D^c} = X_{E^c}\,, \\
(iii) & \quad X_Q = X_{U^c} = X_{D^c} \ \text{and} \ X_L = X_{E^c}\,, \\
(iv) & \quad X_Q = X_{U^c} = X_{E^c} \ \text{and} \ X_L = X_{D^c}\,.
\end{aligned}
\tag{24}
$$

It is straightforward from Eqs. (10,11) that the first two of the above lead to universal $Y_f$ for $f = u, d, e$ and hence identities $(i, ii)$ do not provide realistic description of charged fermion mass hierarchies. Choice $(iii)$ would imply $Y_u \sim Y_d$ and $Y_e \sim Y_\nu$ which is also not in agreement with the observed masses and mixing.

The relation $(iv)$ imposed by AC is similar to the one obtained in $SU(5)$ GUT [15]. In this case, $q_Q = q_{U^c} = q_{D^c} \equiv q_{\mathbf{10}}$, $q_L = q_{D^c} \equiv q_{\bar{\mathbf{5}}}$ with $q_{\mathbf{10}} > q_{\bar{\mathbf{5}}}$ may lead to characteristic features of the quark and lepton masses and mixing angles. Indeed, it has been observed long before that implementation of Froggatt-Nielsen mechanism in $SU(5)$ model lead to a realistic description of the fermion masses [17,19,20]. However, in these models one can choose six independent FN charges for three generations of $\mathbf{10}$ and $\bar{\mathbf{5}}$ if the $U(1)_{\text{FN}}$ is global. In our framework, the AC requirement effectively predict all these six charges in terms of just three parameters: $c$, $q_{\mathbf{10}}$ and $q_{\bar{\mathbf{5}}}$. We show in the next section that while this restriction gives a good understanding of the quark and lepton hierarchies at the leading order, it is not very successful in addressing the detailed quantitative aspects of the observed flavour spectrum. The major limitation comes from the fact that the charges $X_F = q_F(1, 0, -1)$ imply flat profile for the second generation fermions in the fifth dimension. This makes it difficult to explain the hierarchies in masses of the second and third generation fermions.

In order to make all three generations of fermions charged under $U(1)_F$ in an anomaly-free way, at least one of the two conditions in Eqs. (21,22) must be relaxed. Assuming that

Eq. (22) does not hold for all $F$, one finds from Eq. (19) that at least one of the $\mathrm{tr}X_F^3$ must be negative. Fulfilment of AC conditions, Eqs. (19,18), would require specific combinations of inter-generation as well as inter-species $U(1)_F$ charges. In this case, the quark lepton correlations are more complicated and difficult to categorize in an analytical way. It, therefore, requires a systematic numerical analysis of such possibilities for their potential in explaining the flavour hierarchies.

## 4.2   With RH neutrinos

Although the RH neutrinos directly do not contribute in the fermion mass hierarchies obtained from Eqs. (10,13), their presence helps in modifying the AC conditions and enlarging the spectrum of the solutions. For example, one finds a class of solutions characterised by an integer $m$ and

$$\mathrm{tr}X_Q = \mathrm{tr}X_{U^c} = \mathrm{tr}X_{E^c} = m, \quad \mathrm{tr}X_L = \mathrm{tr}X_{D^c} = -3m, \quad \mathrm{tr}X_{N^c} = 5m. \qquad (25)$$

The above choice satisfy all the AC conditions linear in $U(1)_F$. It follows from the fact that the $SU(5)$ representations, $\{Q, U^c, E^c\} \in \mathbf{10}$, $\{L, D^c\} \in \mathbf{\bar{5}}$ and $N^c = \mathbf{1}$, with respective $U(1)_X$ charges $1, -3$ and $5$, can be embedded in an anomaly-free chiral representation of $SO(10)$ [16] which contains $SU(5) \times U(1)_X$ as its subgroup. Further, imposing $SU(5)$ compatible condition $(iv)$ from Eq. (24) and $\mathrm{tr}X_F^3 = \mathrm{tr}X_F$, one can eliminate the remaining $U(1)_Y \times U(1)_F^2$ and $U(1)_F^3$ anomalies, respectively. Similarly, another choice

$$\mathrm{tr}X_Q = \mathrm{tr}X_{D^c} = \mathrm{tr}X_{N^c} = m', \quad \mathrm{tr}X_L = \mathrm{tr}X_{U^c} = -3m', \quad \mathrm{tr}X_{E^c} = 5m', \qquad (26)$$

with integer $m'$ also cancels anomalies involving single $U(1)_F$. The above example follows from embedding of flipped $SU(5)$ [37] and $U(1)_X$ in $SO(10)$. Eqs. (25,26) represent specific examples of more general class of conditions which reduces to Eq. (21) in case of the $U(1)_F$ singlet RH neutrinos. Therefore, the presence of RH neutrinos allows more freedom for anomaly cancellation in the bottom-up approaches.

Several simplified examples with/without RH neutrinos discussed in this section are sufficient to eliminate anomalies. However, it is possible that more complex solutions may exist which cannot be described by the above simplified examples. Such possibilities may imply more subtle correlations among the quark and lepton masses and mixing angles, and it would be worth to investigate them for their ability in explaining the observed flavour spectrum. Therefore, we perform a systematic scan of such possibilities in the next section.

# 5   Numerical search and results

We now perform a numerical scan over anomaly-free $U(1)_F$ charges to investigate their ability to explain the quantitative aspects of the observed hierarchies in the quark and lepton masses and mixings. The system of AC conditions listed in the previous section has been solved following a Diophantine analysis in [38]. The authors of [38] provide a computational algorithm and programme which can lists all possible set of integer $U(1)_F$ charges that can be assigned to the SM fermions and three generations of the RH neutrinos given the maximum absolute charge $|X_{\max}|$. Using this, solutions obtained for only SM fermions for $|X_{\max}| \le 10$ and for the SM fermions along with RH neutrinos for $|X_{\max}| \le 6$ are provided[1] in [38]. For the given $|X_{\max}|$ the number of non-trivial inequivalent solutions with and without RH neutrinos are listed in Table 2.

---

[1] In case of the latter, the authors also list solutions for $7 \le |X_{\max}| \le 10$ in the updated version, see Erratum of [38]. However, we do not use these solutions as we alredy get several viable results for $|X_{\max}| \le 6$.

Table 2: Number of non-trivial distinct solutions for the anomaly-free $U(1)_F$ charges of the SM fermions for $|X_{\max}| \leq 10$, and for SM fermions with three generations of RH neutrinos for $|X_{\max}| \leq 6$ as obtained in [38].

| $|X_{\max}|$ | 1 | 2 | 3 | 4 | 5 | 6 | 7 | 8 | 9 | 10 |
|---|---|---|---|---|---|---|---|---|---|---|
| SM | 7 | 21 | 81 | 250 | 625 | 1982 | 3901 | 7067 | 14353 | 23799 |
| SM + $N^c$ | 37 | 357 | 4115 | 24551 | 111151 | 435304 | - | - | - | - |

We determine the compatibility of each of the solutions for $U(1)_F$ charges with fermion hierarchies in the following way. As it is assumed, the quark and lepton mass hierarchies mainly arise from the elements of $\xi_F$ matrices and the effects of stochastic parameters in $\mathcal{Y}_{u,d,e,\nu}$ can be of $\mathcal{O}(1)$ at most. The physical Yukawa couplings are, therefore, approximated from Eqs. (10,13) as

$$y_{u_i} \simeq \frac{M_c}{\Lambda} \xi_{Q_i} \xi_{U_i^c}, \quad y_{d_i} \simeq \frac{M_c}{\Lambda} \xi_{Q_i} \xi_{D_i^c}, \quad y_{e_i} \simeq \frac{M_c}{\Lambda} \xi_{L_i} \xi_{E_i^c}, \quad y_{\nu_i} \simeq \frac{M_c}{\Lambda} \xi_{L_i}^2. \tag{27}$$

Similarly the mixing angles in the quark and lepton sector are estimated by

$$\theta_{ij}^q \simeq \frac{\xi_{Q_i}}{\xi_{Q_j}}, \quad \theta_{ij}^l \simeq \frac{\xi_{L_i}}{\xi_{L_j}}. \tag{28}$$

Subsequently, we define a $\chi^2$ function

$$\chi^2 = \sum_a \left( \frac{\ln O_a - \ln \bar{O}_a}{\epsilon \ln \bar{O}_a} \right)^2, \tag{29}$$

where $O_a$, $a = 1, 2, ..., 14$ are observable quantities in the flavour sector which include six charged fermion mass ratios and six mixing angles as given in Table 1 along with mass rations $m_b/m_\tau$ and $m_{\nu_2}/m_{\nu_3}$. $\bar{O}_a$ are the corresponding observed values as also listed in Table 1. For the charged fermion mass ratio, we take $\epsilon = 0.1$ while for the mixing angles and neutrino mass ratio we take $\epsilon = 0.5$ as the latter are more sensitive to $\mathcal{O}(1)$ parameters. Note that the above $\chi^2$ does not quantify the absolute deviation of theoretical predictions from the actual experimental data as the exact determination of the observables depends on $\mathcal{O}(1)$ parameters which are not specified yet. It rather provides a measure for a comparative analysis using which the compatibility of various allowed $X_F$ can be quantified. It can be seen that $O_a$, estimated using Eqs. (27,28), do not depend on $M_c/\Lambda$. The $\chi^2$ is therefore a function of only parameter $c$ for the given charges $X_F$ and hence the degree of freedom is $n = 14 - 1 = 13$.

For each set of anomaly-free $U(1)_F$ charges of the SM fermions with/without RH neutrinos from [38], we determine the parameter $c$ by minimizing $\chi^2$. At the minimum, one can approximate order of $\tan\beta \equiv \frac{\langle H_u \rangle}{\langle H_d \rangle}$ from the obtained values of $y_b$, $y_t$ and a relation

$$\tan\beta \simeq \mathcal{O}(1) \frac{y_b}{y_t} \frac{m_t}{m_b}. \tag{30}$$

We consider only fits which give $\tan\beta < 100$. The results of $\chi^2$ minimization are displayed in Table 3 (4) for the case without (with) three generations of RH neutrinos. We list the best fit solution or the solutions with minimised $\chi^2/n \leq 1$ for each $|X_{\max}| \leq 10$ ($|X_{\max}| \leq 6$) in the case without (with) RH neutrinos. For $|X_{\max}| = 5, 6, 7, 8, 10$ in Table 3 and $|X_{\max}| = 4$ in Table 4, we do not find new solution other than already obtained for the smaller $|X_{\max}|$ in

Table 3: The best fit solution for each $|X_{max}| \leq 10$ in the case without RH neutrinos. For $|X_{max}| = 5, 6, 7, 8$ and 10, no new better solution is found.

| $|X_{max}|$ | $\chi^2_{min}/n$ | $c$ | $X_Q$ | $X_{U^c}$ | $X_{D^c}$ | $X_L$ | $X_{E^c}$ |
|---|---|---|---|---|---|---|---|
| 1 | 12.12 | 6.462 | (1,0,-1) | (1,0,-1) | (0,0,0) | (0,0,0) | (1,0,-1) |
| 2 | 8.96 | 2.855 | (1,1,-2) | (2,-1,-1) | (2,-1,-1) | (1,0,-1) | (1,0,-1) |
| 3 | 6.46 | 2.158 | (1,1,-2) | (3,-1,-2) | (1,1,-2) | (1,0,-1) | (3,0,-3) |
| 4 | 4.04 | 1.828 | (2,1,-3) | (4,-1,-3) | (2,0,-2) | (1,0,-1) | (4,0,-4) |
| 9 | 2.29 | 0.962 | (4,3,-7) | (9,-4,-5) | (4,-1,-3) | (1,0,-1) | (8,1,-9) |

Table 4: The best fit solution for each $|X_{max}| \leq 6$ in the case with three generations of RH neutrinos. We also give inequivalent solutions for which $\chi^2_{min}/n \leq 1$. For $|X_{max}| = 4$, no new better solution is found.

| $|X_{max}|$ | $\chi^2_{min}/n$ | $c$ | $X_Q$ | $X_{U^c}$ | $X_{D^c}$ | $X_L$ | $X_{E^c}$ | $X_{N^c}$ |
|---|---|---|---|---|---|---|---|---|
| 1 | 12.12 | 6.462 | (1,0,-1) | (1,0,-1) | (0,0,0) | (0,0,0) | (1,0,-1) | (0,0,0) |
| 2 | 5.34 | 3.765 | (1,1,-1) | (1,-1,-1) | (1,-1,-1) | (0,-1,-2) | (2,1,0) | (2,1,0) |
| 3 | 1.51 | 2.527 | (2,1,-2) | (3,1,-2) | (0,-1,-3) | (0,-1,-2) | (2,1,-3) | (3,3,0) |
| 5 | 0.7 | 2.343 | (2,1,-2) | (2,1,-3) | (1,0,-3) | (0,-1,-2) | (3,1,-2) | (5,0,-1) |
| 5 | 0.89 | 1.619 | (3,2,-3) | (5,1,-3) | (1,-3,-5) | (0,-2,-4) | (4,1,-4) | (5,5,1) |
| 6 | 0.83 | 1.477 | (3,2,-4) | (6,1,-4) | (2,-3,-4) | (0,-1,-2) | (4,1,-6) | (5,2,0) |
| 6 | 0.92 | 1.359 | (4,3,-4) | (4,0,-3) | (-1,-2,-4) | (0,-3,-6) | (5,2,-2) | (6,4,3) |
| 6 | 0.96 | 1.433 | (3,2,-4) | (5,0,-5) | (2,-1,-3) | (0,-1,-2) | (6,1,-5) | (6,2,-4) |
| 6 | 0.97 | 1.242 | (4,3,-5) | (5,-2,-4) | (1,-1,-3) | (0,-2,-4) | (6,2,-3) | (4,2,1) |
| 6 | 0.99 | 1.552 | (2,2,-3) | (6,0,-4) | (2,-1,-5) | (0,-1,-2) | (5,1,-6) | (4,2,0) |

the respective cases. For relative comparison, it may be noted that we obtain $\chi^2_{min} \sim 630$ for $X_F = 0$. The non-zero $U(1)_F$ charges improve the $\chi^2_{min}$ substantially allowing more realistic description of fermion mass hierarchies in the underlying framework.

The noteworthy features of the obtained solutions are:

- All the best fit solutions in Table 3 correspond to $\text{tr} X_F = 0$ for each $F$. The first solution also satisfies Eq. (22) and it is result of $SU(5)$ compatible choice of $X_F$ as discussed in the previous section.

- For the solutions corresponding to $|X_{max}| \geq 2$ in Table 3, the $U(1)_F^3$ AC is arranged by more general condition than Eq. (22). All these solutions restrict second and third generations of $U^c$ to get localised on the SM brane. In this case, the hierarchy between the charm and top mass arise mainly from $X_Q$, and one obtains $m_c/m_t \sim \theta_{23}^q$, which is not in complete agreement with the data.

- The presence of RH neutrinos allow more freedom for anomaly cancellation and does not enforce $\text{tr} X_F = 0$ for the best fit solutions corresponding to $|X_{max}| \geq 2$ as can be seen from Table 4. This leads to considerable improvements in the $\chi^2_{min}$.

- All the solutions with $|X_{max}| \geq 2$ in Table 4 imply second and third generations of $L$ localised on the SM brane and the first generation with a flat profile. These lead to feeble hierarchy in neutrino masses and $\mathcal{O}(1)$ mixing angles in the leptonic sector.

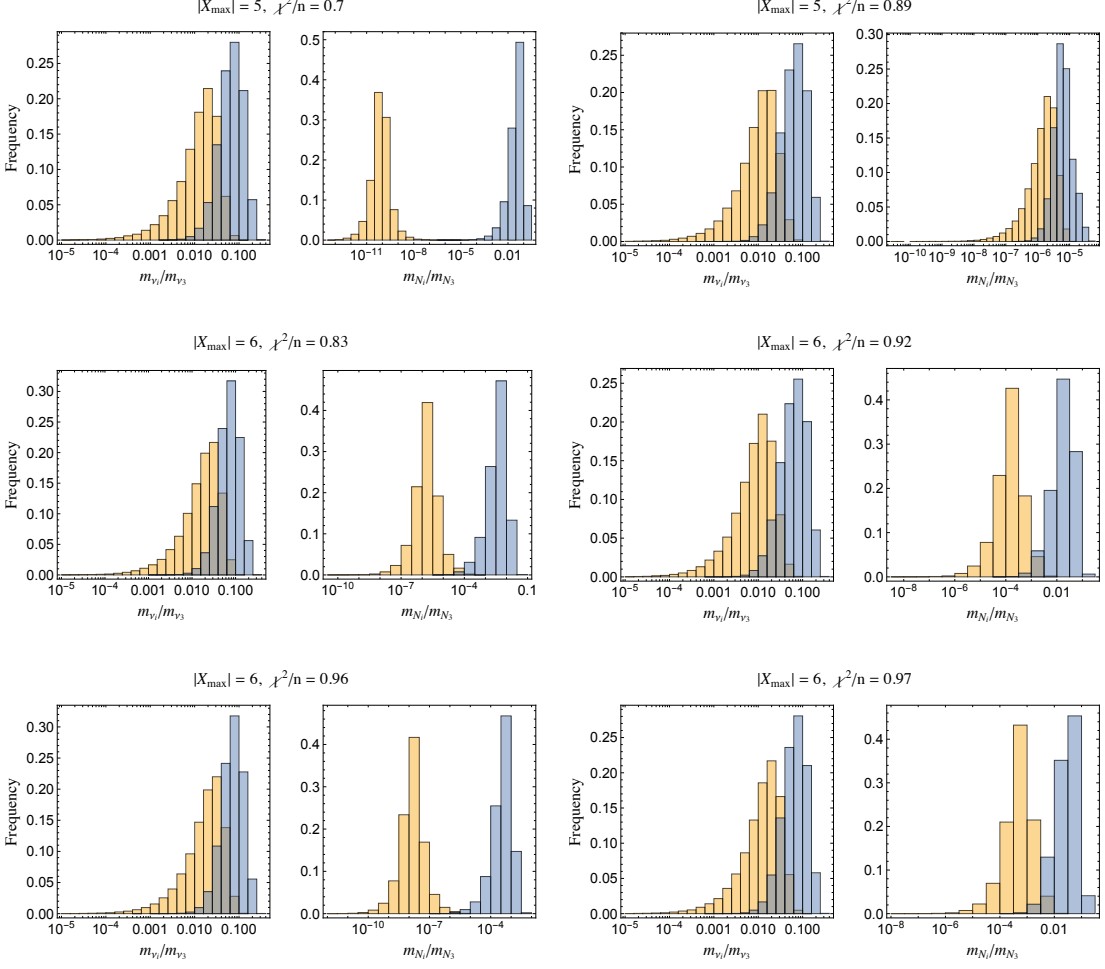

Figure 1: Predictions for the light ($\nu_i$) and heavy ($N_i$) neutrino mass ratios for some of the best fit solutions listed in Table 4. The stochastic parameters are chosen randomly from flat distribution of numbers between 0.5 and 1.

- For most of the solutions corresponding to $\chi^2_{\min}/n \leq 1$, one obtains the first and second generations of $Q$, $U^c$ and $E^c$ localised on the $y = \pi R$ brane while the second and third generations of $D^c$ live very close to the SM brane. These altogether lead to hierarchical charged fermion masses and quark mixing.

From the obtained solutions, predictions for the hierarchies in the light and heavy neutrino masses can be inferred. We do this by choosing the elements of $\mathcal{Y}_\nu$ from a random flat distribution of numbers between 0.1 and 1 and $X_L$ from Table 4 for six best fit solutions. Substituting them back in Eqs. (11,13), we compute the $m_{\nu_1}/m_{\nu_3}$ and $m_{\nu_2}/m_{\nu_3}$. Similar method is followed to determine the RH neutrino mass ratios $m_{N_1}/m_{N_3}$ and $m_{N_2}/m_{N_3}$. The results are displayed in Fig. 1. It can be seen that the RH neutrinos can be extremely hierarchical with masses apart by 5 to 10 orders of magnitude. This arises from the fact that the $U(1)_F$ charges of the RH neutrinos for which the best fit solutions are found are widely separated and different from those of the other matter fields.

To show the compatibility of the best fit solutions with the experimental data, we give an explicit example of $\mathcal{O}(1)$ parameters for the solution corresponding to $\chi^2_{\min}/n = 0.7$ from Table 4 in Appendix A. The elements of $\mathcal{Y}_f$ are determined such that they reproduce the exact values of fermion masses and mixing angles. After all the stochastic parameters are specified, one

can determine $M_c/\Lambda$ and $\tan\beta$ from the absolute values of $m_t$ and $m_b$, respectively. This in turn also allows determination of the absolute mass scale of light as well as heavy neutrinos. The later is linked with the light neutrino mass scale through the type-I seesaw mechanism. The CP violation in both the quark and lepton sectors come entirely from $\mathcal{O}(1)$ parameters and no specific prediction can be made for them. We find that one can obtain their desired values by appropriately choosing $\mathcal{O}(1)$ parameters consistent with the other flavour observables. We show these features for an explicit example given in the Appendix.

For the numerical analysis presented in this paper, we use the fermion masses and mixing data extrapolated at 10 TeV. We do not consider supersymmetric threshold corrections which require complete specification of SUSY breaking sector and scale. However, we expect the results would not change drastically for the other matching scale and/or after inclusion of threshold corrections. Although absolute values of fermion masses are sensitive to such details, the mass ratios and mixing angles we use in the above analysis are mildly sensitive to them. One may expect at most $\mathcal{O}(1)$ effects from these uncertainties which can be adjusted through yet unspecified stochastic parameters.

## 6  Phenomenological Implications

Presence of an extra gauged $U(1)_F$ under which the SM fermions are non-trivially charged implies existence of new gauge interaction for the quarks and leptons. This is mediated by the KK modes of vector field $V_\mu(x, y)$ residing in the vector superfield $\mathcal{V}$. One determines from 5D action, Eq. (1), the following KK expansion:

$$V_\mu(x, y) = \sqrt{\frac{1}{\pi R}}\, Z'_\mu(x) + \sum_{n=1}^{\infty} \sqrt{\frac{2}{\pi R}} \cos\left(\frac{ny}{R}\right) V^n_\mu(x), \tag{31}$$

where the massless mode is identified as $Z'$ boson. $Z'$ can be made massive by introducing a pair of chiral superfield, $\Phi_\pm$ charged under the $U(1)_F$ on $y = 0$ brane. Spontaneous breaking of $U(1)_F$ on the SM brane through the VEVs of scalars in $\Phi_\pm$ then leads to

$$M_{Z'}^2 = g'^2 \left(\langle\phi_+\rangle^2 + \langle\phi_-\rangle^2\right), \tag{32}$$

where $g' = g_5/\sqrt{\pi R}$. The masses of higher KK modes, $V^n_\mu$, are then given by $M_n^2 = M_{Z'}^2 + n^2/R^2$.

The neutral current interactions of the SM fermions with $Z'$ can be obtained from a term in $S_5$:

$$S_{5D} \supset \int_0^{\pi R} dy \int d^4x \int d^4\theta\, \overline{\mathcal{F}}_i e^{2g_5 q\mathcal{V}} \mathcal{F}_i \supset \int d^4x\, g' Z'_\mu X_{F_i}\, \bar{f}_i \gamma^\mu f_i, \tag{33}$$

where $f = q, l, u^c, d^c, e^c$ and $n^c$. The corresponding $U(1)_F$ charges can be read from the respective solutions given in Table 3 or 4. In the physical basis, we obtain

$$\begin{aligned}
X_{F_i}\, \bar{f}_i \gamma^\mu f_i &= (\hat{X}_{u_L})_{ij}\, \overline{u'_{Li}} \gamma^\mu u'_{Lj} + (\hat{X}_{d_L})_{ij}\, \overline{d'_{Li}} \gamma^\mu d'_{Lj} \\
&+ (\hat{X}_{e_L})_{ij}\, \overline{e'_{Li}} \gamma^\mu e'_{Lj} + (\hat{X}_{\nu_L})_{ij}\, \overline{\nu'_{Li}} \gamma^\mu \nu'_{Lj} + L \to R,
\end{aligned} \tag{34}$$

where,

$$\begin{aligned}
\hat{X}_{u_L} &= U_{u_L}^\dagger X_Q U_{u_L}, \quad \hat{X}_{u_R} = -U_{u_R}^\dagger X_{U^c} U_{u_R}, \\
\hat{X}_{d_L} &= U_{d_L}^\dagger X_Q U_{d_L}, \quad \hat{X}_{d_R} = -U_{d_R}^\dagger X_{D^c} U_{d_R}, \\
\hat{X}_{e_L} &= U_{e_L}^\dagger X_L U_{e_L}, \quad \hat{X}_{e_R} = -U_{e_R}^\dagger X_{E^c} U_{e_R}, \\
\hat{X}_{\nu_L} &= U_{\nu_L}^\dagger X_L U_{\nu_L}, \quad \hat{X}_{\nu_R} = -U_{\nu_R}^\dagger X_{N^c} U_{\nu_R}.
\end{aligned} \tag{35}$$

Various matrices $U_f$ appearing in the above relate the flavour basis $f$ with physical basis $f'$ as $f = U_f f'$. They can be determined after the stochastic parameters are fully specified.

Since all the fermions are charged under $U(1)_F$ as required by realsitic fermion mass hierachies as well as the fact that the couplings are flavour non-universal, the mass of $Z'$ is subject to very stringent constraints coming from the direct searches and flavour physics experiments. The strongest direct search constraints come from production of $Z'$ through bottom-quark pair annihilation folllowed by its decay into pair of tau or top quarks at the LHC. Using this, $M_{Z'}$ upto 1.7 (2.2) TeV is excluded by CMS [39] (ATLAS [40]) for $g' \gtrsim 1$. These constraints are more or less indpendent of the flavour structure and mildly depend on the diagonalizing matrices $U_f$ appearing in Eq. (35). More stringent, but flavour structure dependent, limit on $Z'$ comes from $B_s$-$\bar{B}_s$ mixing. Following [41, 42], the current $2\sigma$ limit from $B_s$ mixing implies $M_{Z'} \gtrsim |(\hat{X}_{d_L})_{23}| \times 194$ TeV for $g' \simeq 1$. For an example solution given in the Appendix A, we find $|(\hat{X}_{d_L})_{23}| = 0.13$ and hence $M_{Z'} \gtrsim 26$ TeV. This constraint is however considerably depends on the choice of stochastic parameters which is not unique even for the given set of $U(1)_F$ charges.

Apart from the $Z'$ boson, its higher modes as well as the KK modes of the SM gauge bosons also give rise to tree level flavour changing neutral currents. In this case, the most stringent limit on the compactification scale comes from the contribution of KK gluons in $K$-$\bar{K}$ mixing implying $M_c > \mathcal{O}(10^3)$ TeV [43].

Even though $M_{Z'}$ and $M_c$ are strongly constrained from the various experimental observables, the explanation of fermion mass hierarchies within the proposed framework does not depend on the precise value of $M_{Z'}$ or $M_c$. The parameter which enters in the effective Yukawa couplings is $M_c/\Lambda$, and we obtain $M_c/\Lambda \simeq 10^{-2}$ for the couplings $\mathcal{Y}_f \simeq \mathcal{O}(1)$ which decides the cut-off scale of the theory once the compactification scale is specified. Another independent scale in theory is the scale of $N = 1$ SUSY breaking, namely $M_S$, which can be $\gtrsim \mathcal{O}(10)$ TeV considering various existing constraints on the super-partners of the SM particles. While $M_c$ and $M_S$ can be raised all the way up to the GUT or Planck scale without losing the proposed mechanism of generating flavour hierarchies, their existence at low energies would be desired for stabilization of the electroweak scale.

# 7 Summary

It is well-known that the hierarchical Yukawa couplings in the SM can originate from more fundamental theories with $\mathcal{O}(1)$ couplings constructed in higher spacetime dimension(s). The bulk mass parameter decides localization of massless mode of fermion in the extra dimension and in this way explains the smallness of its Yukawa coupling with the brane localised Higgs field. The bulk mass parameter can be adjusted to get desired coupling in this case and it is possible to explain the observed fermion masses and mixing angles. In this paper, we discuss a framework in which the various bulk mass parameters of the SM fermions are not arbitrary but they arise in a very restrictive manner.

The 5D framework uses supersymmetric gauged $U(1)_F$ symmetry under which the SM fermions and three generations of the so-called right-handed neutrinos are non-trivially charged. Supersymmetry allows only the gauge interactions in the fifth dimension. The bulk mass parameters of all fermions arise from a vacuum expectation value of the $N = 2$ super-partner of $U(1)_F$ gauge field and are proportional to their $U(1)_F$ charges. Orbifold compactification breaks $N = 2$ supersymmetry down to $N = 1$ on the 4D fixed points, one of which hosts the SM gauge and Higgs fields. The requirement from gauge anomaly cancellation severely restricts $U(1)_F$ charges, and in turn predicts correlations between the mass hierarchies of the SM fermions. We discuss such correlations analytically and perform an extensive numerical

search to find solutions compatible with the observed fermion mass spectrum. Several viable solutions are found which are in excellent agreement with the data. These solutions are listed and discussed in detail in section 5.

We find that the RH neutrinos play a significant role in offering anomaly-free solutions for $U(1)_F$ charges of the SM fermions which lead to realistic quark and lepton masses and mixing angles. The $U(1)_F$ charges of RH neutrinos fixed in this way also predict their intergenerational mass hierarchies. It is found that the RH neutrinos can even be more hierarchical than the charged fermions. The model also predicts the existence of $Z'$ boson, which mediates flavour violating interactions in both the quark and lepton sectors in general. However, the mass of $Z'$ and the value of the compactification scale do not depend on fermion mass observables and, therefore, cannot be determined unambiguously. A lower bound on these scales can be put from the direct searches and flavour observables.

# Acknowledgements

We thank Ferruccio Feruglio and Anjan S. Joshipura for a careful reading of the manuscript and valuable suggestions. This work is partially supported by a research grant under INSPIRE Faculty Award (DST/INSPIRE/04/2015/000508) from the Department of Science and Technology, Government of India. The computational work reported in this paper was performed on the High Performance Computing (HPC) resources (Vikram-100 HPC cluster) at the Physical Research Laboratory, Ahmedabad.

# A   Stochastic parameters for the best fit solution

We give an explicit example of values of $\mathcal{O}(1)$ parameters in $\mathcal{Y}_f$, $f = u, d, e, \nu$, which reproduce the realistic fermion mass spectrum. For the best fit solution corresponding to $\chi^2 = 0.7$ in Table 4, the $\xi_F$ matrices, as defined in Eq. (11), are obtained as:

$$
\begin{aligned}
\xi_Q &= \text{Diag.}\left(2.824 \times 10^{-2}, 0.209, 3.062\right), \\
\xi_{U^c} &= \text{Diag.}\left(2.824 \times 10^{-2}, 0.209, 3.749\right), \\
\xi_{D^c} &= \text{Diag.}(0.209, 1.0, 3.749), \\
\xi_L &= \text{Diag.}(1.0, 2.175, 3.062), \\
\xi_{E^c} &= \text{Diag.}\left(3.321 \times 10^{-3}, 0.209, 3.062\right), \\
\xi_{N^c} &= \text{Diag.}\left(3.955 \times 10^{-5}, 1.0, 2.175\right).
\end{aligned}
\tag{36}
$$

For the above, we optimize the stochastic parameters, with constraint $0.1 \leq |(\mathcal{Y}_f)_{ij}| \leq 1$, such that they reproduce the observed fermion mass spectrum. We also assume symmetric $\mathcal{Y}_f$ for

simplicity. In this way, the determined values of these parameters are:

$$\mathcal{Y}_u = \begin{pmatrix} -0.3249 + 0.3661i & 0.3394 + 0.2392i & 0.15 + 0.1778i \\ 0.3394 + 0.2392i & -0.646 - 0.0005i & 0.023 - 0.1268i \\ 0.15 + 0.1778i & 0.023 - 0.1268i & -0.3089 + 0.6353i \end{pmatrix},$$

$$\mathcal{Y}_d = \begin{pmatrix} -0.0993 + 0.0174i & 0.0815 + 0.1363i & 0.0579 + 0.0946i \\ 0.0815 + 0.1363i & 0.1609 - 0.0722i & -0.1 - 0.0008i \\ 0.0579 + 0.0946i & -0.1 - 0.0008i & -0.005 + 0.149i \end{pmatrix},$$

$$\mathcal{Y}_e = \begin{pmatrix} -0.0991 - 0.0763i & 0.0581 + 0.1073i & -0.1025 + 0.0155i \\ 0.0581 + 0.1073i & -0.2084 - 0.0017i & -0.1017 + 0.0808i \\ -0.1025 + 0.0155i & -0.1017 + 0.0808i & -0.1218 + 0.0076i \end{pmatrix},$$

$$\mathcal{Y}_\nu = \begin{pmatrix} -0.8802 - 0.4746i & -0.0651 + 0.7028i & 0.2849 + 0.5176i \\ -0.0651 + 0.7028i & -0.4633 + 0.0007i & -0.6934 + 0.3762i \\ 0.2849 + 0.5176i & -0.6934 + 0.3762i & 0.1486 + 0.6189i \end{pmatrix}. \qquad (37)$$

The above values when substituted in Eqs. (10,13) reproduces the *exact central values* of the charged fermion mass ratios, quark and lepton mixing angles as listed in Table 1 and solar and atmospheric neutrino squared mass differences as listed in [28] for the normal ordering. The CP violating phases in the quark and lepton sector are obtained as $\delta_{\text{CKM}} = 1.208$ and $\delta_{\text{PMNS}} = -0.262$, respectively which are in agreement with the current global fits.

Specification of stochastic parameters allows one to compute $\tan\beta$, $M_c/\Lambda$ from $m_b$ and $m_t$ and to estimate $\Lambda'$ from the atmospheric neutrino scale. $\Lambda'$ determines the mass of the lightest neutrino in this setup. These are obtained as

$$\tan\beta = 13.9, \quad \frac{M_c}{\Lambda} = 0.098, \quad \Lambda' = 5.7 \times 10^{14}\,\text{GeV}, \quad m_{\nu_1} = 0.008\,\text{eV}, \qquad (38)$$

where we use $\langle H_u \rangle^2 + \langle H_d \rangle^2 = (174\,\text{GeV})^2$. Note that the above predictions are sensitive to the exact values of $\mathcal{O}(1)$ parameters. They vary for different choice of stochastic parameters even for the fixed $U(1)_F$ charges and $c$.

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
