# Peer review of "Fermion mass hierarchies from supersymmetric gauged flavour symmetry in 5D"

_SciPost Physics, doi:SciPost Phys. 10, 154 (2021)_

## Round 2 · Referee Report · Anonymous (Referee 1) · 2021-6-12

Strengths

Tha main advantage of this framework over the traditional four dimensional supersymmetric FN models comes from the following: It decouples the fermion Yukawa couplings and the constraints from anomaly cancellation conditions. This significantly provides more parameter space especially when one includes masses and mixing angles of neutral leptons.

Weaknesses

The main phenomenological issue which could be the issue is with flavour violating constraints especially in the charged leptonic sector. The strong constraints from EWPT could also be an issue in these models. A few comments in this direction would be good.

Report

The present work proposes a flavour model based on a supersymmetric gauged U(1) model in 5 dimensions compactified on S1/Z2.The extra-dimensional space is considered to have flat geometry with vector and hyper fields propagating in the bulk. Higgs is localised on the IR brane. Fermions zero modes attain their masses through localisation in the bulk. The profiles In the bulk are determined by their bulk masses which are in turn determined by U(1) charges. The model also considers the lepton sector, including neutrino masses. They are included through the Weinberg operator. The model is chiral and the cancellation of the anomalies of the zero modes leads to an explanation of the features of the fermion masses like hierarchies in masses, mixing angles in the which have been listed in the introduction.
The paper is well written , well explained and is numerically sound.

  • validity: high
  • significance: high
  • originality: top
  • clarity: top
  • formatting: perfect
  • grammar: excellent

Author:  Ketan Patel  on 2021-06-14  [id 1504]

(in reply to Report 1 on 2021-06-12)
Category:
remark

I am very thankful to the referee(s) for their constructive report on this paper. I would like to comment on the issue concerning the flavour violation and Electro-Weak Precision Tests (EWPT) as raised by the referee(s).

In the proposed framework, the new physics relevant for flavour violation and EWPT arises from the Z' boson and its Kaluza-Klein(KK) excitations. Unfortunately, the model does not predict the mass scales of these fields and they have to be fixed from the experimental constraints. As we have discussed in section VI in the paper, the present limits from flavour changing neutral currents already push the masses of these new excitations in multi (tens and hundreds of) TeV regime. Also, such a heavy spectrum of Z' boson and its KK excitations is not in conflict with the present constraints from EWPT.

I look forward to hearing from the referee(s) and/or from the editor(s) if they have any comment/suggestion in this regard.

Thank you.

Sincerely,
Ketan M. Patel

---

## Editorial Decision

published